# Mobilization of multilineage-differentiating stress-enduring cells into the peripheral blood in liver surgery

**Koji Kikuchi**[1]*, **Hirokatsu Katagiri**[1], **Yuji Suzuki**[2,3,4], **Hiroyuki Nitta**[1], **Akira Sasaki**[1]

**1** Department of Surgery, Iwate Medical University School of Medicine, Yahaba, Iwate, Japan, **2** Division of Hepatology, Department of Internal Medicine, Iwate Medical University School of Medicine, Yahaba, Iwate, Japan, **3** Institute for Biomedical Sciences Molecular Pathophysiology, Iwate Medical University, Yahaba, Iwate, Japan, **4** Division of Allergy and Rheumatology, Department of Internal Medicine, Iwate Medical University School of Medicine, Yahaba, Iwate, Japan

* k0jikikuchi@yahoo.co.jp

**Data Availability Statement:** All relevant data are within the paper and its Supporting Information files.

## Abstract

### Purpose

This study investigated whether liver damage severity relates to the mobilization of multilineage-differentiating stress-enduring (Muse) cells, which are endogenous reparative pluripotent stem cells, into the peripheral blood (PB) and whether the degree of mobilization relates to the recovery of liver volume following human liver surgery.

### Methods

Forty-seven patients who underwent liver surgery were included in the present study. PB-Muse cells were counted before surgery, on postoperative days (PODs) 3 and on POD 7. Liver volume was measured using computed tomography before and after surgery.

### Results

The PB-Muse cell count increased after surgery. The number of PB-Muse cells before surgery was higher, but without statistical significance in the group with neoplasms than in the healthy group that included liver donors ($p = 0.065$). Forty-seven patients who underwent liver surgery were divided into major hepatic resection (MHR; hepatectomy of three or more segments according to the Couinaud classification, n = 22) and minor hepatic resection (mhr; hepatectomy of two segments or less according to the Couinaud classification, n = 25) groups. PB-Muse cells increased at high rates among MHR patients ($p = 0.033$). Except for complication cases, PB-Muse cells increased at higher rates in the group with advanced liver volume recovery ($p = 0.043$). The predictive impact of the rate of increase in PB-Muse cells on the recovery of liver volume was demonstrated by multivariate analysis (OR 11.0, $p = 0.014$).

### Conclusions

PB-Muse cell mobilization correlated with the volume of liver resection, suggesting that the PB-Muse cell number reflects the degree of liver injury. Given that the degree of PB-Muse

**Funding:** Japan Society for the Promotion of Science, JP17K10676, Hirokatsu Katagiri Japan Society for the Promotion of Science, JP18K15825, Yuji Suzuki Takeda Science Foundation, Yuji Suzuki.

**Competing interests:** I have read the journal's policy and the authors of this manuscript have the following competing interests: Yuji Suzuki received a research grant from Takeda Science Foundation.

cell mobilization was related to liver volume recovery, PB-Muse cells were suggested to contribute to liver regeneration, although this mechanism remains unclear.

## Introduction

Liver surgery is associated with problems such as prolonged severe liver dysfunction after massive hepatectomy and postoperative liver failure. None of these conditions has infallible treatments. Preclinical studies using fatal cirrhosis and massive hepatectomy swine, rat, and mouse models have examined mesenchymal stem cell (MSC) transplantation to improve survival rates and liver function. Cell transplantation as a regenerative treatment is expected to eventually complement existing therapies for liver dysfunction [1].

Multilineage-differentiating stress-enduring (Muse) cells are endogenous reparative pluripotent stem cells that are stress tolerant [2]. They express genes relevant to pluripotency, such as Oct3/4, Nanog, and Sox2; can be identified as pluripotency surface marker stage specific embryonic antigen (SSEA)-3-positive cells in the bone marrow, peripheral blood (PB), and connective tissues of various organs; able to differentiate into triploblastic cells and self-renew at a single cell level while they are non-tumorigenic and exhibit low telomerase activity, consistent with the fat that they normally reside in the body [3–7]. Muse cells in the circulating blood can selectively home to damaged tissue by sensing sphingosine-1-phosphate (S1P), one of the general signals of tissue injury produced by phosphorylating the cell membrane component sphingosine in damaged cells, using S1P receptor 2 [8]. After homing to damaged tissue, Muse cells replace damaged/apoptotic cells by spontaneously differentiating into the same cell types as the damaged/apoptotic cells and repairing the tissue [8–11]. Therefore, they do not require gene introduction or differentiation induction for rendering pluripotency or a differentiated state. Furthermore, allogenic Muse cells can be used without human leucocyte antigen (HLA) matching or long-term immunosuppressant treatment, partly due to HLA-G expression implicated in immunotolerance of the placenta [8]. Clinical trials in which donor-derived Muse cells are delivered by intravenous infusion without HLA-matching and immunosuppressant treatment to patients with acute myocardial infarction, stroke, epidermolysis bullosa, spinal cord injury, neonatal cerebral palsy, and amyotrophic lateral sclerosis are in progress, and their safety and efficacy have been reported in acute myocardial infarction and epidermolysis bullosa [12, 13].

Endogenous Muse cells may act as reparative stem cells through the above-described mechanisms. Suppose that extensive tissue damage is caused, the post-infarct tissue produces S1P as a damage alert signal, thereby mobilizing endogenous Muse cells, probably from the bone marrow to the circulating blood; the Muse cells then home to the damaged tissue and repair the tissue. Clinical data support this hypothesis because the number of PB-Muse cells increases after the onset of stroke and acute myocardial infarction [14, 15]. In acute myocardial infarction, patients with a high number of PB-Muse cells in the acute phase exhibit statistically meaningful cardiac function recovery, with less occurrence of heart failure at 6 months [15]. Thus, the number of endogenous Muse cells is a potential parameter of reparative activity.

Previous research on a physical hepatectomy mouse model showed that Muse cells accumulate only in the vicinity of damaged tissue, differentiating into multiple cell types that comprise the liver [16]. Similarly, data from an acute hepatitis mouse model confirmed that Muse cells migrate to damaged livers and differentiate into hepatocyte marker-positive cells [17]. This study investigated the dynamics of Muse cells to examine whether their mobilization is related to the severity of tissue damage and contributes to the recovery of liver volume after liver surgery in humans.

## Materials and methods

### Definitions

Based on the Couinaud classification, we defined major hepatic resection (MHR) as a hepatectomy of three or more segments, and minor hepatic resection (mhr) as a hepatectomy of two segments or less. Max PB-Muse was defined number of PB-Muse cells on postoperative day (POD) 3 or POD7, whichever was larger. Max AST was defined maximum aspartate aminotransferase on PODs 3 and 7. We defined the rate of increase in the PB-Muse cell count (Max PB-Muse /PB-Muse cell number before surgery) as ΔMuse, and the rate of increase in liver volume (liver volume on POD 7/ expected remnant liver volume after surgery) as ΔVolume. We graded postoperative morbidity according to the Clavien–Dindo classification [18], and postoperative mortality as any death occurring within 90 days of liver surgery.

### Study protocol and patients

This prospective study recruited patients who were admitted to Iwate Medical University Hospital from October 2020 to August 2021 and underwent liver surgery. No exclusion criteria were used. The Ethics Committee of Iwate Medical University Hospital approved the study (reference MH2018-038). All patients provided their written informed consent before the study commenced, and the investigation conformed to the principles of the Declaration of Helsinki.

### SSEA-3-positive Muse cell numbers in peripheral blood

The Muse cell number in the PB was measured based on SSEA3$^+$ cell counts [7]. To quantify circulating SSEA3$^+$ cells, 10 ml of PB was obtained per patient at each of three time points (before surgery and on PODs 3 and 7). Mononuclear cells were isolated by Lymphoprep™ (STEMCELL Technologies, Vancouver, BC, Canada) treatment. Anti-SSEA-3 antibody (1:100; BioLegend, San Diego, CA, USA) was added and incubated for 60 min in the dark at 4˚C with gentle mixing. The cells were then washed three times with phosphate-buffered saline. Next, fluorescein isothiocyanate (FITC)-labeled anti-rat IgM antibody (1:200; Jackson Immunoresearch, West Grove, PA, USA) was added as a secondary antibody, and the cells were incubated and washed using the same method as the primary antibody. Finally, NucRed™ Live 647 (ThermoFisher, Waltham, MA, USA) was added for nuclear staining, and the cells were incubated for 15 min at room temperature. FACS (FACS Aria II™; Becton Dickinson, San Jose, CA, USA) and FACSDiva™ (Becton Dickinson) software were used for the analysis. Unincubated cells with no antibody, cells incubated with only secondary antibody (FITC-labeled anti-rat IgM antibody), and cells incubated with rat IgM (1:100; BioLegend) followed by the secondary antibody (FITC-labeled anti-rat IgM antibody) as an isotype control were used to determine nonspecific reactions and autofluorescence, as well as to set the gate for the SSEA-3$^+$ cells [7].

### Blood biochemical analysis

Blood samples underwent complete blood cell count and biochemical analyses for the determination of aspartate aminotransferase (AST), alanine aminotransferase (ALT), total bilirubin (T-Bil), and C-reactive protein (CRP) concentration and prothrombin time-international normalized ratio (PT-INR).

### Liver volumetry

Computed tomography (CT) using a Synapse Vincent analyzer (Fujifilm Medical, Tokyo, Japan) was performed before and seven days after surgery to calculate liver volume. The

expected remnant liver volume after surgery was estimated using preoperative CT measurements, and ΔVolume was calculated. A study on post-hepatectomy liver volume in 538 patients found that the greatest amount of liver regeneration occurred on POD 7. Also, the ΔVolume was ≥1.19 on POD 7 when more than 10% of the liver was resected [19]. Based on this, 1.19 was selected as the cut-off value for ΔVolume, and patients were divided into two groups: those with sufficient liver volume recovery (n = 22) and those with insufficient recovery (n = 25).

## Statistical analysis

Data are presented as means ± standard deviations (SDs), and categorical variables are described as totals and frequencies. Data greater than +/- three times the SDs of the number of PB-Muse cells before surgery and Max PB-Muse were excluded as outliers. Differences in groups were assessed using the Mann-Whitney U test for continuous variables and a chi-squared or Fisher's exact test (for expected counts of <5) for categorical variables. Correlations were analyzed using Spearman's rank correlation. We performed multivariate logistic regression analysis using variables that had $p$-values <0.05 in the univariate analysis. The Youden index was used to determine the optimal cutoff value for the receiver operating characteristic curve (ROC). Statistical analysis was performed using JMP software version 14.2.0 (SAS Institute, Cary, North Carolina). Variables with $p$-values <0.05 were considered statistically significant.

## Results

### Characteristics and clinical features

This prospective study enrolled 59 patients who underwent liver surgery. Twelve cases were excluded because they were inappropriate for the study (one outlier case, five non-resection cases, and six data-deficient cases). Of the 47 patients finally included in the present study, 22 underwent MHR, and 25 underwent mhr. A significant difference was found between in the two groups in the number of HCC patients (Table 1). Surgical approach, operation time, blood loss, maximum white blood cells on PODs 3 and 7, maximum T-Bil on PODs 3 and 7, and length of hospital stay were significantly higher in the MHR group (Table 2). PB-Muse cells were also measured in 10 healthy subjects.

### Muse cell numbers in peripheral blood

Fig 1 shows the representative results of the quantification of Muse cells using FACS. The mononuclear cell fraction after Lymphoprep treatment was roughly selected by forward and side scatter gating (Fig 1A). The few remaining red blood cells, which were negative for NucRed Live 647, were removed by specific gravity centrifugation (Fig 1B). Nonspecifically labeled cells were assessed on the basis of the isotype control (Fig 1C). Finally, gating was set for the SSEA-3+ cells (Fig 1D). In each FACS analysis, the nonspecific bindings were assessed using isotype control, followed by the setting of the gate for the SSEA-3+ cells. The number of PB-Muse cells was expressed as cells per 100 μL of blood, as follows: the number of Muse cells (/100μL) = mononuclear cells (/100μL) × SSEA3$^+$ cells (%).

We measured the number of PB-Muse cells before surgery in 59 cases and in an additional 10 healthy subjects (age, 29.6 ± 5.13 years; male-to-female ratio, 9:1). Among the 69 subjects, one case was excluded as an outlier (n = 68). The number of PB-Muse cells was higher, but without statistical significance, in the group with both benign and malignant neoplasms (426 ± 454.5 cells/100μL, n = 54) than in the healthy group, which included four liver donors (131.7 ± 160.2 cells/100μL, n = 14, $p$ = 0.065; Fig 2A). In the group with both benign and

**Table 1. Baseline characteristics.**

| | MHR (n = 22) | mhr (n = 25) | *p*-value |
|---|---|---|---|
| Age (years) | 63.7 ± 14.6 | 66.4 ± 12.9 | 0.281 |
| Sex | | | 0.627 |
| Male | 18 (81.8%) | 19 (76.0%) | |
| Female | 4 (18.2%) | 6 (24.0%) | |
| BMI (kg/m$^2$) | 23.5 ± 3.1 | 25.2 ± 4.9 | 0.138 |
| Smoking | 2 (9.1%) | 7 (28.0%) | 0.100 |
| Alcohol intake | 9 (40.9%) | 8 (32.0%) | 0.526 |
| Chemotherapy | 5 (22.7%) | 9 (36.0%) | 0.321 |
| ASA | | | 0.439 |
| I | 10 (45.5%) | 7 (28.0%) | |
| II | 8 (36.4%) | 13 (52.0%) | |
| III | 4 (18.2%) | 5 (20.0%) | |
| CCI | 3.0 ± 2.7 | 2.2 ± 2.2 | 0.409 |
| C-P class | | | 0.926 |
| A | 21 (95.5%) | 24 (96.0%) | |
| B | 1 (4.6%) | 1 (4.0%) | |
| HV | | | 0.159 |
| HBV | 1 (4.6%) | 4 (16.0%) | |
| HCV | 1 (4.6%) | 4 (16.0%) | |
| WBC (× 10$^3$/μL) | 5.7 ± 1.6 | 6.1 ± 2.7 | 0.725 |
| AST (U/L) | 26.7 ± 15.2 | 28.9 ± 14.9 | 0.331 |
| ALT (U/L) | 25.7 ± 13.2 | 29.9 ± 22.6 | 0.907 |
| T-Bil (mg/dL) | 0.6 ± 0.3 | 0.6 ± 0.3 | 0.855 |
| PT-INR | 1.1 ± 0.1 | 1.1 ± 0.1 | 0.529 |
| CRP (mg/dL) | 0.7 ± 1.3 | 1.1 ± 2.2 | 0.438 |
| Diagnosis | | | |
| HCC | 2 (9.1%) | 11 (45.8%) | 0.008 |
| Metastasis | 8 (36.4%) | 8 (33.3%) | 0.753 |
| Other (malignant) | 8 (36.4%) | 5 (20.8%) | 0.211 |
| Benign | 4(18.2%) | 1 (4.0%) | 0.116 |
| Tumor diameter (mm) | 47.6 ± 37.3 | 32.4 ± 26.9 | 0.093 |

Values are expressed as means ± SDs.

MHR, major hepatic resection; mhr, minor hepatic resection; BMI, body mass index; ASA, American Society of Anesthesiologists; CCI, Charlson comorbidity index; C-P, Child-Pugh score; HV, hepatitis virus; AST, aspartate aminotransferase; WBC, white blood cells; ALT, alanine aminotransferase; T-Bil, total bilirubin; PT-INR, prothrombin time-international normalized ratio; CRP, C-reactive protein; HCC, hepatocellular carcinoma.

malignant neoplasms (n = 54), the number of PB-Muse cells did not differ between the presence (415.1 ± 367.9 cells/100μL) and absence (434 ± 515.3 cells/100μL) of metastasis (*p* = 0.558, Fig 2B).

The number of PB-Muse cells increased after surgery (before surgery 394.8 ± 458.8 cells/100μL, POD3 971.4 ± 1306.5 cells/100μL, POD7 645.9 ± 881.2 cells/100μL, Fig 3A). Within 7 days of surgery, five cases had grade 3 or higher complications according to the Clavien–Dindo classification (complication cases). In these cases, the number of PB-Muse cells did not differ before and after surgery (before surgery 416.4 ± 525.1 cells/100μL, POD3 193.6 ± 230.3 cells/100μL, POD7 220.8 ± 227.3 cells/100μL, Fig 3B). Except for the complication cases (no-complication cases), the number of PB-Muse cells increased and peaked on POD 3 (before

**Table 2. Comparison of surgical outcomes after major hepatic resection and minor hepatic resection.**

| | MHR (n = 22) | mhr (n = 25) | *p*-value |
|---|---|---|---|
| Surgical approach | | | 0.038 |
| LLR | 15 (68.2%) | 23 (92.0%) | |
| Open | 7 (31.8%) | 2 (8.0%) | |
| Multiple hepatectomy | 4 (18.2%) | 5 (20%) | 0.874 |
| Operation time (min) | 280.1 ± 70.0 | 187.9 ± 66.9 | 0.001 |
| Pringle maneuver (min) | 54.6 ± 23.7 | 45.6 ± 43.1 | 0.397 |
| Blood loss (ml) | 279.2 ± 321.5 | 108.2 ± 153.7 | 0.002 |
| Blood transfusion | 1 (4.8%) | 0 (0.0%) | 0.270 |
| Max. WBC (× $10^3$/μL) | 10.3 ± 2.5 | 9.2 ± 3.5 | 0.044 |
| Max. AST (U/L) | 472.5 ± 211.2 | 415.8 ± 276.1 | 0.228 |
| Max. ALT (U/L) | 412.8 ± 184.6 | 344.1 ± 228.3 | 0.186 |
| Max. T-Bil (mg/dL) | 2.0 ± 1.3 | 1.2 ± 0.7 | 0.005 |
| Max. PT-INR | 1.3 ± 0.1 | 1.2 ± 0.1 | 0.337 |
| Max. CRP (mg/dL) | 8.1 ± 5.7 | 10.2 ± 6.2 | 0.233 |
| Morbidity Within seven days of surgery (Clavien-Dindo ≥ grade III) | 3 (13.6%) | 2 (8.0%) | 0.532 |
| Bile leak | 1 (4.5%) | 1 (4.0%) | |
| Pancreatic fistula | 1 (4.5%) | 0 (0.0%) | |
| Pseudoaneurysm | 1 (4.5%) | 0 (0.0%) | |
| Heart failure | 0 (0.0%) | 1 (4.0%) | |
| Length of hospital stay (days) | 19.4 ± 14.6 | 13.7 ± 10.5 | 0.004 |
| Mortality | 1 (4.6%) | 0 (0.0%) | 0.291 |

Values are expressed as means ± SDs.

MHR, major hepatic resection; mhr, minor hepatic resection; LLR, laparoscopic liver resection; max. WBC, maximum white blood cells on PODs 3 and 7; max. AST, maximum aspartate aminotransferase on PODs 3 and 7; max. ALT, maximum alanine aminotransferase on PODs 3 and 7; max. T-Bil, maximum total bilirubin on PODs 3 and 7; max. PT, maximum prothrombin time on PODs 3 and 7; max. CRP, maximum C-reactive protein on PODs 3 and 7.

surgery 392.2 ± 457.3 cells/100μL, POD3 1063.9 ± 1351.8 cells/100μL, POD7 696.5 ± 917.4 cells/100μL, Fig 3B). The rate of change from POD 3 to 7 (PB-Muse cell number on POD 7/ PB-Muse cell number on POD 3) was higher, but without statistical significance in the complication cases (3.1 ± 5.5) than in the no-complication cases (1.1 ± 1.8, *p* = 0.168; Fig 3C).

## Comparison of MHR and mhr

The number of PB-Muse cells before surgery did not significantly differ between the MHR (351.1 ± 456.5 cells/100μL) and mhr groups (433.2 ± 466.6 cells/100μL; Fig 4A). The Max PB-Muse was higher, but without statistical significance, in the MHR group (1,458.1 ± 1,567.9 cells/100μL) than in the mhr group (847.4 ± 1,047.9 cells/100μL, *p* = 0.079; Fig 4B). The ΔMuse was significantly higher in the MHR group (17.4 ± 33.9) than in the mhr group (4.6 ± 8.8, *p* = 0.033; Fig 4C).

## Liver volumetry and Muse cell numbers

Of the 47 patients, 22 had ΔVolume ≥ 1.19, and 25 had ΔVolume < 1.19. ΔMuse did not differ between the two groups (ΔVolume ≥ 1.19 13.0 ± 22.9, ΔVolume < 1.19 8.4± 26.4, *p* = 0.241, Fig 5A), however, at the verification stage, except for the five cases with complications according to the Clavien–Dindo classification (grade 3 or higher within 7 days of surgery), ΔMuse was significantly higher in patients with ΔVolume ≥ 1.19 (16.3 ± 25.2) than in those with ΔVolume <1.19 (8.4 ± 26.4, *p* = 0.043; Fig 5B).

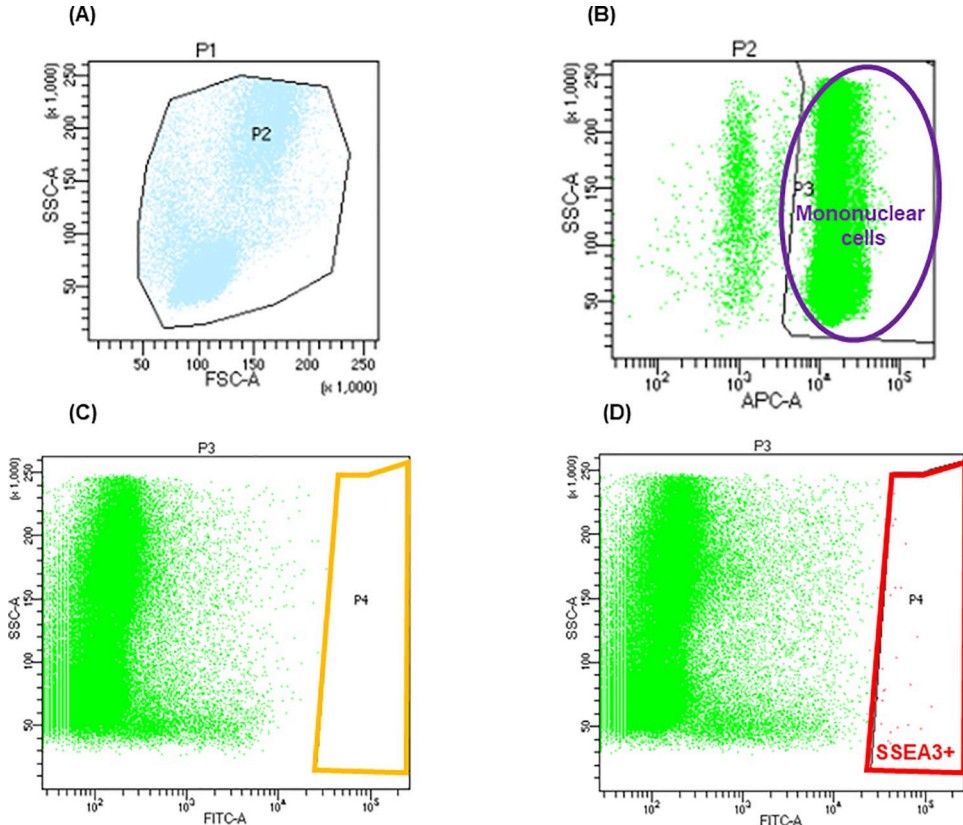

**Fig 1. How to count SSEA3 positive Muse cells using fluorescence-activated cell sorting (FACS).** A: The mononuclear cell fraction after Lymphoprep™ treatment was roughly selected by forward scatter and side scatter gating. B: The few remaining red blood cells (negative for NucRed™ Live647) were removed by specific gravity centrifugation. C: Nonspecifically labeled cells were removed based on the isotype control. D: The gating was set for SSEA-3$^+$ cells.

### Factors affecting ΔVolume

In the univariate analysis, many of the factors relating to ΔVolume were compared between the ΔVolume < 1.19 and ΔVolume ≥ 1.19 groups for the no-complication cases (Table 3). Among these factors, ΔMuse was significantly higher in the ΔVolume ≥ 1.19 group than in the ΔVolume < 1.19 group ($p = 0.043$). Significant differences in surgical approach were found between the two groups ($p = <0.001$; Table 3); therefore, we performed multivariate logistic regression analysis including these two factors. The ROC curve analysis of ΔMuse with respect to ΔVolume identified 6.0 as the optimal cut-off point [area under the ROC curve (AUC) 0.687, 95% confidence interval (CI) 0.514–0.859; Fig 6], with 52.9% sensitivity and 88.0% specificity. Even after adjusting for the surgical approach, ΔMuse was significantly higher in the ΔVolume ≥ 1.19 group than in the ΔVolume < 1.19 group (odds ratio 11.0, 95% CI 1.63–74.05, $p = 0.014$; Table 4).

## Discussion

In this study, we evaluated the PB-Muse cell dynamics involved in human liver surgery. To evaluate the severity of tissue damage, we divided the 47 patients into the MHR and mhr groups. In the MHR group, as expected, operation time, blood loss, and length of hospital stay were significantly higher than in the mhr group (Table 2). Therefore, we regarded the MHR

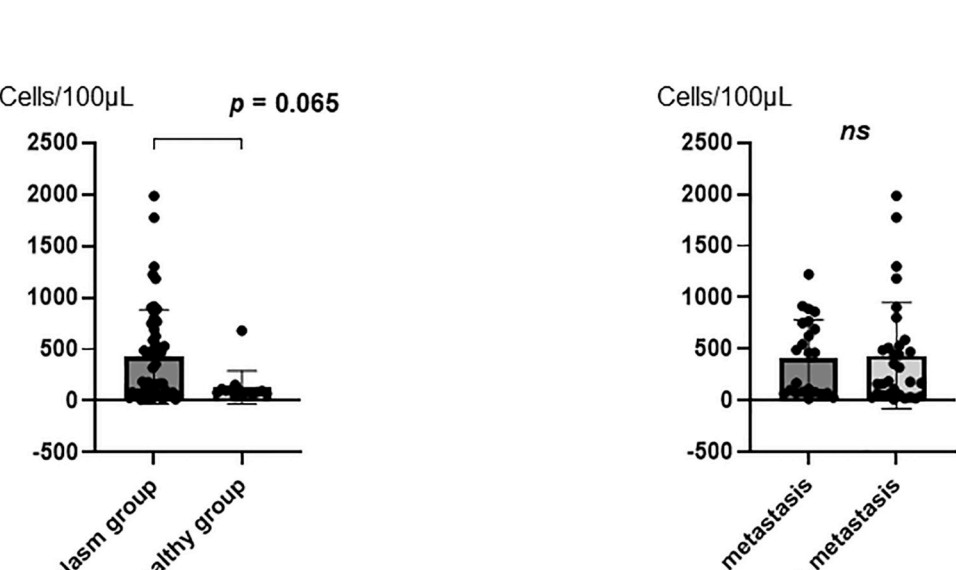

**Fig 2. The number of PB-Muse cells before surgery.** A: Comparison of neoplasm group and healthy group. B: Comparison of presence and absent of metastasis.

group as a more invasive group. Preoperative PB-Muse cell number varied considerably from case to case. We found that the number of preoperative PB-Muse cells was higher, but without statistical significance, in the group with both benign and malignant neoplasms ($p = 0.065$; Fig 2A). This outcome suggests that the presence of neoplasms affects preoperative PB-Muse cell

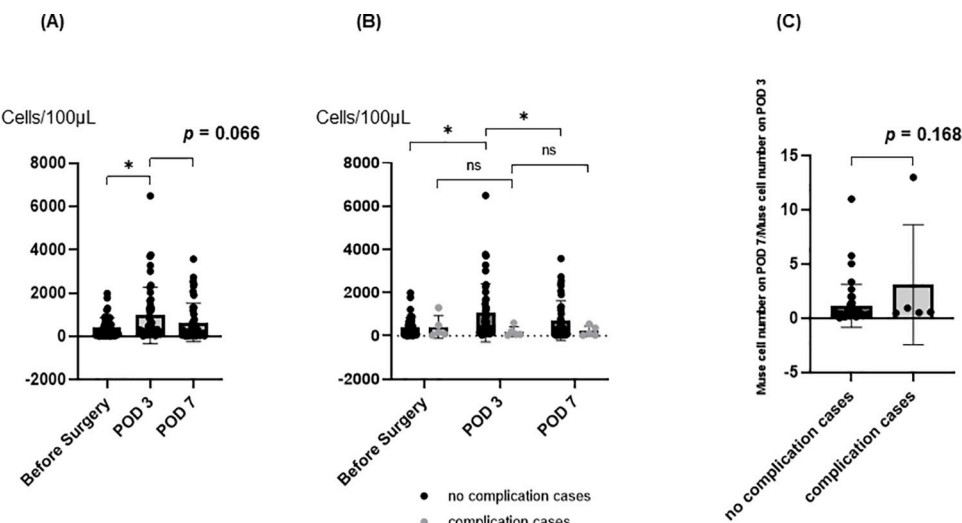

**Fig 3. Muse cell numbers in peripheral blood.** A: Changes in the Muse cell number in peripheral circulating blood after hepatectomy. B: Changes in the Muse cell number in peripheral circulating blood after hepatectomy for complication and no-complication cases. C: The rate of PB-Muse cell number changes from postoperative days (PODs) 3 to 7 (PB-Muse cell number on POD 7/PB-Muse cell number on POD 3) in complication and no-complication cases. $^{*}p < 0.05$.

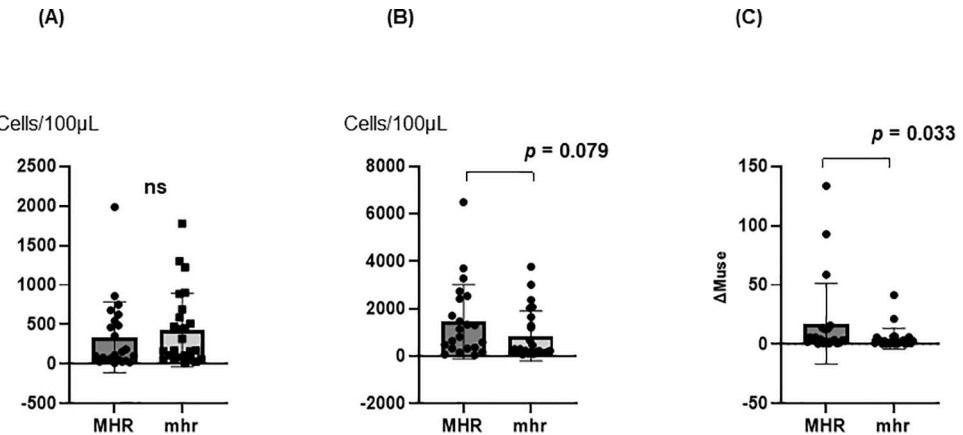

**Fig 4. Comparison of MHR and mhr.** A: The number of PB-Muse cells before surgery. B: Max PB-Muse in the MHR and mhr groups. C: ΔMuse in the MHR groups and mhr group. Max. Muse = maximum PB-Muse cell number on PODs 3 and 7.

number. Although smoking or drinking habits have been reported to influence Muse cell dynamics [14], our study showed no differences in preoperative PB-Muse cell number based on both traits (S1 and S2 Tables).

Our study revealed that PB-Muse cell mobilization was significantly higher after surgery than before-surgery (Fig 3A), Hence, PB-Muse cells were mobilized by surgical invasion. In the complication cases, the rate of change in the PB-Muse cell number from POD 3 to 7 was higher, but without statistical significance (Fig 3C). We also found that ΔMuse was significantly higher in the MHR group than in the mhr group (Fig 4C). These results demonstrate that PB-Muse cell mobilization depends on the resected liver volume and may also be triggered in response to complicating infections or circulatory disorders that cause tissue damage.

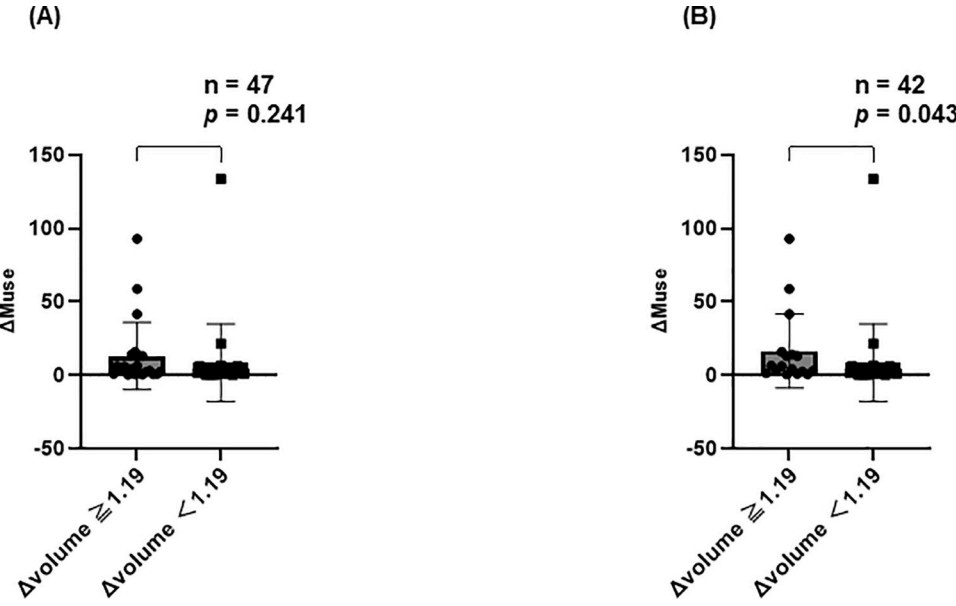

**Fig 5. Liver volumetry and Muse cell numbers.** A: Comparison of ΔMuse between the Δvolume ≥1.19 and the Δvolume <1.19 groups. B: Comparison of ΔMuse between the Δvolume ≥1.19 and the Δvolume <1.19 groups for the no-complication cases.

**Table 3. Comparison of the factors affecting Δ volume in no complication cases.**

| | Δvolume ≧ 1.19 (n = 17) | Δvolume < 1.19 (n = 25) | P-value |
|---|---|---|---|
| Age (years) | 62.1 ± 13.4 | 65.9 ± 13.4 | 0.323 |
| Sex | | | 0.482 |
| Male | 12 (70.6%) | 20 (80.0%) | |
| Female | 5 (29.4%) | 5 (20.0%) | |
| BMI, (kg/m$^2$) | 24.8 ± 3.4 | 24.4 ± 5.1 | 0.591 |
| Smoking | 2 (11.8%) | 6 (24.0%) | 0.322 |
| Alcohol intake | 4 (23.5%) | 9 (36.0%) | 0.391 |
| DM | 6 (35.3%) | 7 (28.0%) | 0.616 |
| Liver cirrhosis | 1 (5.9%) | 0 (0.0%) | 0.220 |
| Chemotherapy | 6 (35.3%) | 8 (32.0%) | 0.824 |
| C-P class | | | 0.232 |
| A | 17 (100.0%) | 23 (92.0%) | |
| B | 0 (0.0%) | 2 (8.0%) | |
| Surgical approach | | | < 0.001 |
| MHR | 13 (76.5%) | 6 (24.0%) | |
| mhr | 4 (23.5%) | 19 (76.0%) | |
| Max. Muse (cells/100μL) | 1425.5 ± 1695.9 | 1109.0 ± 1134.0 | 0.530 |
| ΔMuse | 16.3 ± 25.2 | 8.4 ± 26.4 | 0.043 |
| Pringle maneuver (min) | 54.7 ± 25.1 | 47.0 ± 41.1 | 0.661 |

Values are expressed as means ± SD.

BMI, body mass index; DM, diabetes mellitus; C-P, Child–Pugh score; MHR, major hepatic resection; mhr, minor hepatic resection; max. Muse, maximum PB-Muse cell number on PODs 3 and 7.

Because Muse cell accumulation is specific to damaged tissue, this characteristic could be useful for objectively evaluating surgical invasion and quantifying damage severity.

S1P is a migration factor for Muse cells in patients with myocardial infarction [15]. We hypothesized that S1P would also act as a Muse cell migration factor following a hepatectomy. In the present study, the plasma S1P concentrations of 15 patients were analyzed; however, in our study, S1P did not exhibit the same dynamics as those reported for myocardial infarction in humans or other animals (S1 Appendix and S1 Fig). Previous studies have revealed the involvement of S1P in the proliferation, motility, morphology, and differentiation of tumor cells, neurons, vascular smooth muscle cells, and vascular endothelial cells; the protein is associated with five specific G protein-coupled receptors (S1PR1–5) [20–22]. In a study using a myocardial infarction rabbit model, S1PR2 expression was higher in Muse cells than in non-Muse cells. Inhibiting S1PR2 reduced the number of Muse cells engrafted in the left ventricle, increased the infarct range, and reduced the recovery of the left ventricular ejection fraction [8]. In vascular endothelial cells and bone marrow-derived cells, S1PR2 has an inhibitory effect on tumor angiogenesis and growth [23]. One confounding factor in our study was that 11 of the 15 patients (73.3%) who underwent S1P measurements had malignant disease; thus, our results may not exclusively reflect the effects of hepatectomy. A different experimental design and larger sample size will be needed to clarify the relationship between S1P and Muse cell mobilization in liver tissue damage.

ΔMuse was significantly different between the ΔVolume < 1.19 and ΔVolume ≥ 1.19 groups for the no-complication cases, even after adjustment for surgical approach (odds ratio 11.0, 95% CI 1.63–74.05, $p$ = 0.014; Table 4). Liver regeneration after hepatectomy occurs primarily via hepatocyte hypertrophy [24]; however, the details of this mechanism and the

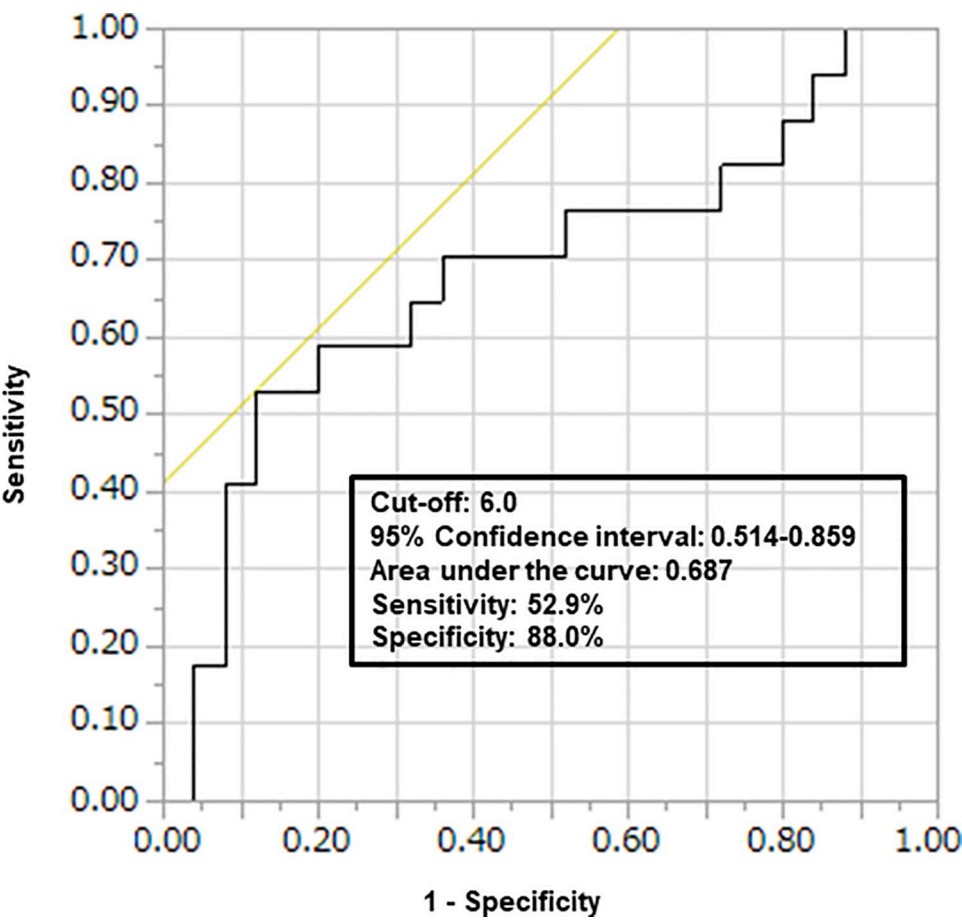

**Fig 6. Receiver operating characteristic curve analysis of the ΔMuse versus the Δvolume.**

potential involvement of non-liver cells remain poorly understood. In a study of a 70% hepatectomy swine model, administering porcine Muse cells via the portal vein improved jaundice and PT-INR [11]. Administering human bone marrow-derived MSCs and human bone marrow-derived Muse cells to a small-for-size rat model after liver transplantation showed that Muse cells migrated to the transplanted liver, and Ki-67-positive hepatocytes and sinusoidal endothelial cells were significantly higher in the Muse cell group. Furthermore, this study found that Muse cells could protect sinusoidal endothelial cells by expressing hepatocyte growth factor and vascular endothelial growth factor-A [25]. These studies suggest that Muse cells contribute to promoting liver function and regeneration. In the present study, in addition to the resected liver volume, known as a factor in liver volume recovery, the increase in PB-Muse cells may also be associated with post-hepatectomy liver volume recovery, suggesting a contribution to liver regeneration; however, this study could not verify the detailed mechanism of their involvement. A previous study using hepatectomy mouse models revealed that

**Table 4. Multivariate logistic regression analysis of factors associated with Δ volume.**

| Variables | Odds ratio | 95% Confidence interval | P-value |
|---|---|---|---|
| Surgical approach (MHR or mhr) | 13.0 | 2.30–73.54 | 0.004 |
| ΔMuse (cut-off: 6.0) | 11.0 | 1.63–74.05 | 0.014 |

exogenously administered human Muse cells migrate and engraft to the injured liver and injured site, where they spontaneously differentiate into liver components, including hepatocytes; however, the frequency of human–mouse fusion cells is very low [16]. The role of Muse cells in liver volume recovery is likely to be functional, promoting tissue repair and liver regeneration rather than quantitative supplementation as trophic-like effects of MSCs [26].

The study had several limitations. First, the sample size may have been too small to statistically analyze many complex factors. Second, human clinical research lacks an established technique for labeling bone marrow-derived Muse cells, meaning that we could not perform detailed assessments of Muse cell engraftment, differentiation, and proliferation in liver tissue. Third, factors such as limitations on hospital stays meant that we could not improve the reliability of our conclusions by conducting a longer follow-up study. Fourth, we were not able to evaluate the relationship between PB-Muse cells and liver function because no fatal complications occurred during the measurement period, and only limited cases exhibited reduced liver function. Studies using animal models would provide further opportunities to establish a direct causal link.

In conclusion, PB-Muse cells are mobilized based on the resected liver volume, the degree of PB-Muse cell mobilization was related to liver volume recovery. Their concentration in PB can potentially be used to quantify the degree of surgical invasion. Furthermore, Muse cell mobilization could contribute to liver regeneration, although the detail of this mechanism remains poorly understood.

## Supporting information

**S1 Appendix. Plasma S1P concentration.**
(DOCX)

**S1 Fig. Plasma S1P concentrations and Muse cell numbers.** A: Changes in the number of PB-Muse cells after hepatectomy. B: Changes in plasma S1P concentrations after surgery. C: Plasma S1P levels (maximum S1P levels on PODs 1, 3, and 7) in the MHR and mhr groups. D: Relationship between plasma S1P levels and the Max PB-Muse. S1P = sphingosine-1-phosphate. $^*p < 0.05$.
(TIF)

**S1 Table. Comparison of the preoperative PB-Muse cell numbers affecting smoking.**
(DOCX)

**S2 Table. Comparison of the preoperative PB-Muse cell numbers affecting drinking.**
(DOCX)

**S3 Table. Effect size of analysis with significant difference.**
(DOCX)

## Acknowledgments

We are grateful to Drs. Mari Dezawa, Yasuhiro Takikawa, Takeshi Iwaya, and Keisuke Kakisaka for helpful discussions. We thank Drs. Syoji Kanno, Akira Umemura, Daiki Takeda, Hayato Nagase, Kenji Makabe, Satoshi Amano, Shingo Yanari, and Taku Kimura for their collaboration on this work.

## Author Contributions

**Methodology:** Hirokatsu Katagiri, Yuji Suzuki, Hiroyuki Nitta.

**Supervision:** Hirokatsu Katagiri, Akira Sasaki.

**Writing – original draft:** Koji Kikuchi.

**Writing – review & editing:** Koji Kikuchi.

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
