## [Decision Letter · Decision Letter 0]

28 Mar 2022

PONE-D-22-04405Mobilization of multilineage-differentiating stress-enduring cells into the peripheral blood in liver surgery: A cohort studyPLOS ONE

Dear Dr. kikuchi,

Thank you for submitting your manuscript to PLOS ONE. After careful consideration, we feel that it has merit but does not fully meet PLOS ONE’s publication criteria as it currently stands. Therefore, we invite you to submit a revised version of the manuscript that addresses the points raised during the review process.

We look forward to receiving your revised manuscript.

Kind regards,

Chengming Fan, MD, PhD

Academic Editor

PLOS ONE

Journal Requirements:

[I have read the journal's policy and the authors of this manuscript have the following competing interests:

Yuji Suzuki received a research grant from Takeda Science Foundation.]

Additional Editor Comments:

Please response to the reviewers point by point.

Reviewers' comments:

Reviewer's Responses to Questions

**Comments to the Author**

1. Is the manuscript technically sound, and do the data support the conclusions?

Reviewer #1: Yes

Reviewer #2: Yes

2. Has the statistical analysis been performed appropriately and rigorously? 

Reviewer #1: I Don't Know

Reviewer #2: I Don't Know

3. Have the authors made all data underlying the findings in their manuscript fully available?

Reviewer #1: Yes

Reviewer #2: Yes

4. Is the manuscript presented in an intelligible fashion and written in standard English?

Reviewer #1: Yes

Reviewer #2: Yes

5. Review Comments to the Author

Reviewer #1: In this article, Kikuchi et al. demonstrated the number of Muse cells in the peripheral blood is elevated after the liver surgery compared to the before surgery, particularly in patients with liver tumors and those who underwent major hepatic resection. The results reflect that the increase in Muse cells in the peripheral blood after the liver damage may facilitate liver repair, even though Muse cell differentiation and proliferation mechanisms are unclear.

Even the study is interesting, there are many concerns:

1. In blood loss, Max, T-Bil, Length of hospital stay (days) of table 2, figure 2A, 3A, 3B, 4B, 4C, 5A and 6B, the authors described that there are significant statistical differences. However, there seems to be no significant difference, since standard deviation is bigger than mean value. How the authors conducted statistical analysis? A column-scatter plot is recommended to reflect the distribution.

2. In diagnosis table 1 and surgical approach table 3, the authors described p-value. Please show which group there are significant difference.

3. Figure 1C: the gate p4 should be moved to the right to exclude a non-specific dot. The numbers of positive Muse cells should be calculated based on the newly set gate. Did the authors strictly set the gate according to the criteria of the FACS gate setting? The authors should check all the gate settings and re-analyse.

4. The authors define the number of peripheral blood Muse cells as follows; “The number of PB-Muse cells was expressed as cells per 100 ul of blood, as follows: number of PB-Muse cells (/100 ul)=mononuclear cells (/100 ul)×SSEA-3+ cells (%).” Have the authors investigated whether there is a change in the overall number of white blood cells due to liver resection or tumor formation?　 These basic data need to be added to the result section.

5. The authors discuss the study of S1P concentration in peripheral blood that “One confounding factor in our study was that 11 of the 15 patients (73.3%) who underwent S1P measurements had malignant disease; thus, our results may not exclusively reflect the effects of hepatectomy.”　Did the authors investigate whether there is a difference between S1P levels before hepatectomy and S1P levels in healthy subjects? If there is a difference, it would be one of the reasons for this discussion, so please add the data if you have it.

6. Basic information (age, gender, etc.) for healthy subjects is not listed. Please add this information to the results section.

7. Figure 2A: the authors mentioned “n=69” in figure 2A. According to the main text, the sample numbers of neoplasm are 55, and healthy numbers are 14. The “n=69” is too confusing and can be misinterpreted as 69 neoplasm samples plus 69 healthy samples used for analysis.

8. Although the authors showed in figure 3C that the ratio of POD7/POD3 is higher in complication cases, how to explain on POD3 in figure 3B that the Muse number is lower in complication cases than no complication cases?

9. p14 line1, what is the unit of “Δvolume”?

10. p15 line1, although table 2 is mentioned, "tumor diameter" is shown in table 1.

11. Regarding the same "tumor diameter", the p-value is shown as 0.06. Is this significant?

Reviewer #2: Muse cells are pluripotent non-tumorigenic cells with high capacity of tissue regeneration and function recovery. In addition, Muse cells are endogenous cells present in connective tissues (i.e. bone marrow, adipose tissue) that go into circulation (PB-Muse cells) to repair damage tissue when an injury occurs as it was demonstrated in patients undergoing cerebral stroke and acute myocardial infarction.

In this manuscript Kikuchi et al., analyzed Muse cells mobilization into the peripheral blood in patients with liver tumors that underwent major or minor hepatic resection (MHR and mhr, respectively) at days 3 and 7 post-surgery. The number of circulating Muse cells significantly increased in the MHR vs mhr patients. In addition, there was a positive correlation between circulating Muse cells and recovery of liver volume after surgery.

This is an interesting paper, with novel findings which could have implications in the process of liver regeneration after hepatectomy through the mobilization and repair capacity of endogenous Muse cells.

There are some issues that need to be clarified:

1- Fig. 3D and 3E show values of a patient with max Muse cell>10,000 and delta Muse >120 that look to be outliers. How was the liver recovery and AST values of this particular patient before and at days 3 and 7 after surgery?

2- Remove Fig 5 because of lack of power (only n=15). Include these results in Discussion as data not shown.

3- If S1P is not a major player in mobilization of Muse cells, what other mechanism(s) could be proposed?

Minor changes:

1- The terms maximum PB-Muse cell number and maximum AST are quite confusing, both need to be replaced.

2- Include in the reference section paper by Heneidi et al, 2013.

6. PLOS authors have the option to publish the peer review history of their article (what does this mean?). If published, this will include your full peer review and any attached files.

Reviewer #1: No

Reviewer #2: No

---

## [Author Response · Author response to Decision Letter 0]

15 May 2022

Dr. Emily Chenette 

Editor-in-Chief

Dr. Chengming Fan

Academic Editor

PLOS ONE

May 16, 2022

PONE-D-22-04405

Mobilization of multilineage-differentiating stress-enduring cells into the peripheral blood in liver surgery.

Thank you very much for the opportunity to revise our manuscript. The reviewer's comments were very helpful. We have corrected the manuscript point-by-point and have provided responses to each comment below. We believe that the manuscript has been substantially improved, and we would be grateful if you would consider its suitability for publication in PLOS ONE. We look forward to hearing from you.

Sincerely,

Koji Kikuchi M.D.

Department of Surgery, 

Iwate Medical University School of Medicine, 

2-1-1 Idai-dori, Yahaba, Iwate, 028-3609, Japan

Response to Editor 

Comments to the Author

We note that you have included the phrase “data not shown” in your manuscript. Unfortunately, this does not meet our data sharing requirements. PLOS does not permit references to inaccessible data. We require that authors provide all relevant data within the paper, Supporting Information files, or in an acceptable, public repository. Please add a citation to support this phrase or upload the data that corresponds with these findings to a stable repository (such as Figshare or Dryad) and provide and URLs, DOIs, or accession numbers that may be used to access these data. Or, if the data are not a core part of the research being presented in your study, we ask that you remove the phrase that refers to these data.

Reply: 

Thank you very much for your suggestion. We have added this data to the Supporting Information files (S1 and S2 Tables).

Response to Reviewer #1:

Comments to the Author

In this article, Kikuchi et al. demonstrated the number of Muse cells in the peripheral blood is elevated after the liver surgery compared to the before surgery, particularly in patients with liver tumors and those who underwent major hepatic resection. The results reflect that the increase in Muse cells in the peripheral blood after the liver damage may facilitate liver repair, even though Muse cell differentiation and proliferation mechanisms are unclear.

Even the study is interesting, there are many concerns

1. In blood loss, Max, T-Bil, Length of hospital stay (days) of table 2, figure 2A, 3A, 3B, 4B, 4C, 5A and 6B, the authors described that there are significant statistical differences. However, there seems to be no significant difference, since standard deviation is bigger than mean value. How the authors conducted statistical analysis? A column-scatter plot is recommended to reflect the distribution.

Reply: 

Thank you for your valuable comments. In blood loss, Max, T-Bil, Length of hospital stay (days) of table 2, figure 2A, 3A, 3B, 4B, 4C, 5A and 6B, differences in groups were assessed using the Mann-Whitney U test. The effect size of each analysis is shown in the Supporting Information file (S3 Table). The figures have been changed to a column scatter plot (Figs. 2A, 2B, 3A, 3B, 3C, 4A, 4B, 4C, 5A, 5B, S1A, S1B, and S1C).

2. In diagnosis table 1 and surgical approach table 3, the authors described p-value. Please show which group there are significant difference.

Reply: 

Thank you for pointing this out. In Table 1, significantly more patients with HCC belonged to the mhr group. As you suggested, the description that was difficult to understand has been changed (Page 9, Lines 27). In Table 3, significantly more patients underwent MHR in the ΔVolume ≥ 1.19 group (Page 14, Lines 35).

3. Figure 1C: the gate p4 should be moved to the right to exclude a non-specific dot. The numbers of positive Muse cells should be calculated based on the newly set gate. Did the authors strictly set the gate according to the criteria of the FACS gate setting? The authors should check all the gate settings and re-analyse.

Reply: 

Thank you very much for your invaluable comments. In each FACS analysis, the number of positive Muse cells was used to calculate the newly set gate on the basis of the isotype control. This information has been added in the revised manuscript (Page 10, Lines 36). We have checked all the gate settings, performed a reanalysis for some patients, and changed the figure (Fig. 1C). 

4. The authors define the number of peripheral blood Muse cells as follows; “The number of PB-Muse cells was expressed as cells per 100 ul of blood, as follows: number of PB-Muse cells (/100 ul)=mononuclear cells (/100 ul)×SSEA-3+ cells (%).” Have the authors investigated whether there is a change in the overall number of white blood cells due to liver resection or tumor formation?　 These basic data need to be added to the result section.

Reply: 

In accordance with your suggestion, we have added text and data on the change in the overall number of white blood cells in the Results section (Page 8, Lines 29), Table 1 (Page 9, Lines 20) and Table 2 (Page 10, Lines 13).

5. The authors discuss the study of S1P concentration in peripheral blood that “One confounding factor in our study was that 11 of the 15 patients (73.3%) who underwent S1P measurements had malignant disease; thus, our results may not exclusively reflect the effects of hepatectomy.”　Did the authors investigate whether there is a difference between S1P levels before hepatectomy and S1P levels in healthy subjects? If there is a difference, it would be one of the reasons for this discussion, so please add the data if you have it.

Reply: 

Thank you for providing these insights. In the present study, we did not analyze S1P concentrations in healthy subjects. As you pointed out, if the S1P levels significantly differed between the pre-hepatectomy and healthy subjects, I think it is possible to consider something. This will be a subject for future research. 

6. Basic information (age, gender, etc.) for healthy subjects is not listed. Please add this information to the results section.

Reply: 

As suggested by the reviewer, we have added this data to the results section (Page 11, Lines 17).

7. Figure 2A: the authors mentioned “n=69” in figure 2A. According to the main text, the sample numbers of neoplasm are 55, and healthy numbers are 14. The “n=69” is too confusing and can be misinterpreted as 69 neoplasm samples plus 69 healthy samples used for analysis.

Reply: 

We agree with your assessment. In Figure 2, the descriptions of n = 69 and n = 55 have been removed (Figs. 2A and B).

8. Although the authors showed in figure 3C that the ratio of POD7/POD3 is higher in complication cases, how to explain on POD3 in figure 3B that the Muse number is lower in complication cases than no complication cases?

Reply: 

Thank you very much for your invaluable comments. As the number of complicated cases was as small as 6, it is difficult to give a definitive explanation. The complications were bile leak (onset at 2POD), bile leak (4POD), bile leak (7POD), pancreatic fistula (7POD), heart failure (3POD), and pseudoaneurysm (5POD). Assuming that Muse cell mobilization is suppressed at the onset of complications, this can be explained by the small number of Muse cells in complicated cases. In the process of repairing damaged tissue, the dynamics of Muse cells against further invasion such as complications is also an issue for future study.

9. p14 line1, what is the unit of “Δvolume”?

Reply: 

Thank you for pointing this out. Since ΔVolume is a ratio, there is no unit like ΔMuse.

10. p15 line1, although table 2 is mentioned, "tumor diameter" is shown in table 1.

Reply: 

Thank you for your comment. As you pointed out, this is a mistake. The content has been corrected (Page 8, Lines 27).

11. Regarding the same "tumor diameter", the p-value is shown as 0.06. Is this significant?

Reply: 

Thank you for your comment. As you pointed out, this is a mistake. This is not significant. The content has been corrected (Page 8, Lines 27).

Response to Reviewer #2:

Comments to the Author

Muse cells are pluripotent non-tumorigenic cells with high capacity of tissue regeneration and function recovery. In addition, Muse cells are endogenous cells present in connective tissues (i.e. bone marrow, adipose tissue) that go into circulation (PB-Muse cells) to repair damage tissue when an injury occurs as it was demonstrated in patients undergoing cerebral stroke and acute myocardial infarction.

In this manuscript Kikuchi et al., analyzed Muse cells mobilization into the peripheral blood in patients with liver tumors that underwent major or minor hepatic resection (MHR and mhr, respectively) at days 3 and 7 post-surgery. The number of circulating Muse cells significantly increased in the MHR vs mhr patients. In addition, there was a positive correlation between circulating Muse cells and recovery of liver volume after surgery.

This is an interesting paper, with novel findings which could have implications in the process of liver regeneration after hepatectomy through the mobilization and repair capacity of endogenous Muse cells.

There are some issues that need to be clarified:

1- Fig. 3D and 3E show values of a patient with max Muse cell>10,000 and delta Muse >120 that look to be outliers. How was the liver recovery and AST values of this particular patient before and at days 3 and 7 after surgery?

Reply: 

We thank the referee for the valuable comment. The patient underwent percutaneous transhepatic abscess drainage for liver abscess three times in the posterior segment before surgery. A follow-up CT scan showed that the abscess in the posterior segment had been replaced with a mass, and the patient was diagnosed with liver metastasis and underwent posterior segmentectomy. The ΔVolume in this patient was 0.93. The AST values were 26 (before surgery), 1200 (POD3), and 62 U/L (POD7).

2- Remove Fig 5 because of lack of power (only n=15). Include these results in Discussion as data not shown.

Reply: 

Thank you very much for your important comments. We agree with you, and Fig. 5 has been removed because of the lack of power. However, this is the first report of the dynamics of S1P in liver surgery, and we consider it to be valuable data, despite its lack of power. Thus, we will show the data in the Supporting Information files (S1 Appendix and S1 Fig).

3- If S1P is not a major player in mobilization of Muse cells, what other mechanism(s) could be proposed?

Reply: 

We appreciate the reviewer's comment on this point. No mechanism can be proposed at this time. The SDF-1-CXCR4 axis may also be a candidate factor that triggers Muse cell mobilization. As a result of animal experiments, however, it is said that the S1P-S1PR2 axis is more specific to Muse cell mobilization. Owing to the design of this study, it may be difficult for the number of Muse cells and S1P concentrations to increase. Although many animal experiments use fatal disease and complication models, in liver surgery, we determine the transection line to avoid congestion or ischemia in the remnant liver. That is, we perform surgery to minimize tissue damage. For this reason, it is possible that the mobilization of Muse cells and the S1P concentration were not noticeable in this study.

Minor changes:

1- The terms maximum PB-Muse cell number and maximum AST are quite confusing, both need to be replaced.

Reply: 

In accordance with your suggestion, we have replaced these terms with shorter terms. Max PB-Muse was defined as the number of PB-Muse cells on postoperative day (POD) 3 or POD7, whichever was larger. Max AST was defined as the maximum aspartate aminotransferase on PODs 3 and 7. This information has been added in the revised manuscript (Page 5, Lines 25.)

2- Include in the reference section paper by Heneidi et al, 2013.

Reply: 

As suggested by the reviewer, we have added this paper to the reference section (Page 20, Lines 20).

---

## [Decision Letter · Decision Letter 1]

7 Jun 2022

PONE-D-22-04405R1Mobilization of multilineage-differentiating stress-enduring cells into the peripheral blood in liver surgery: A cohort studyPLOS ONE

Dear Dr. koji kikuchi,

Thank you for submitting your manuscript to PLOS ONE. After careful consideration, we feel that it has merit but does not fully meet PLOS ONE’s publication criteria as it currently stands. Therefore, we invite you to submit a revised version of the manuscript that addresses the points raised during the review process.

We look forward to receiving your revised manuscript.

Kind regards,

Chengming Fan, MD, PhD

Academic Editor

PLOS ONE

Journal Requirements:

Additional Editor Comments:

Please response to the reviewers' comments point by point.

Reviewers' comments:

Reviewer's Responses to Questions

**Comments to the Author**

1. If the authors have adequately addressed your comments raised in a previous round of review and you feel that this manuscript is now acceptable for publication, you may indicate that here to bypass the “Comments to the Author” section, enter your conflict of interest statement in the “Confidential to Editor” section, and submit your "Accept" recommendation.

Reviewer #1: (No Response)

Reviewer #2: All comments have been addressed

2. Is the manuscript technically sound, and do the data support the conclusions?

Reviewer #1: Yes

Reviewer #2: Partly

3. Has the statistical analysis been performed appropriately and rigorously? 

Reviewer #1: I Don't Know

Reviewer #2: I Don't Know

4. Have the authors made all data underlying the findings in their manuscript fully available?

Reviewer #1: Yes

Reviewer #2: Yes

5. Is the manuscript presented in an intelligible fashion and written in standard English?

Reviewer #1: Yes

Reviewer #2: Yes

6. Review Comments to the Author

Reviewer #1: The authors improved the manuscript. However, there still several concerns to be cleared before publication.

1. Since bar graphs were changed to plots, now the data became interpretable. On the other hand, values for average and SD overlapped with plots in the revised figures. For this, the referee asks authors to describe numerical values of average and SD in the main text, so that readers may understand the meaning of data. For example, page 11, second and third paragraphs, sentences for Fig 2 and Fig 3 needs numerical values for each group, neoplasm and healthy groups.

2. looking at Fig 2A and 2B, one single plot in neoplasm group seem to elevate the average in neoplasm group. They stated in the text that “the number of PB-Muse cell was higher in the group with both benign and malignant neoplasms (n =55) than in the healthy group, which included four liver donors (n = 14, p = 0.056; Fig. 2A).” Since there was no statistical difference, authors should not carelessly state higher. “High but without statistical significance” might be better sentence.

The same issue falls into other “higher” parts throughout the text.

Furthermore, authors should check the appropriateness of the single dot in the neoplastic/metastatic patients in Fig 2 to be included as a proper sample. This single dot seems to distorts the data that it should be.

3. page 11, third paragraph: why authors can conclude that POD3 was peak in Fig 3A. There was no statistical significance between POD3 and POD7 (p=ns). The same problem falls onto Fig 3B, as well. How POD7 can be the highest while POD3 and POD7 were ns.

Again a single plot in Fig 3C seems to be extraordinal case and thus is questionable to be included as one of the rational samples.

4. page 13, top line: the MHR group (1,852.0 ± 2,431.8 cells/100μL) was higher than in the mhr group (847.4 ± 1,047.9 cells/100μL, p = 0.051; Fig. 4B -> higher but without statistical significance.

5. page 11: “ΔMuse was significantly higher in patients with ΔVolume ≥ 1.19 (16.3 ± 25.2) than in those with ΔVolume ˂1.19 (8.4 ± 26.4, p = 0.043; Fig. 5B). ΔMuse did not correlate with ΔVolume (p = 0.917; Fig. 5C).” -> The two sentences are inconsistent. Needs to be explained or reconsider the statement.

6. Table 2 is sloppy. Extra lines should be deleted.

Reviewer #2: Figures 3D/E have to be redone without including the outlier. P values needs to be re-calculated and results and discussion needs to modified accordingly.

7. PLOS authors have the option to publish the peer review history of their article (what does this mean?). If published, this will include your full peer review and any attached files.

Reviewer #1: No

Reviewer #2: No

---

## [Author Response · Author response to Decision Letter 1]

29 Jun 2022

Dr. Emily Chenette 

Editor-in-Chief

Dr. Chengming Fan

Academic Editor

PLOS ONE

June 29, 2022

PONE-D-22-04405

Mobilization of multilineage-differentiating stress-enduring cells into the peripheral blood in liver surgery.

Thank you very much for the opportunity to revise our manuscript. The reviewer's comments were very helpful. We have corrected the manuscript point-by-point and have provided responses to each comment below. We believe that the manuscript has been substantially improved, and we would be grateful if you would consider its suitability for publication in PLOS ONE. We look forward to hearing from you.

Sincerely,

Koji Kikuchi M.D.

Department of Surgery, 

Iwate Medical University School of Medicine, 

2-1-1 Idai-dori, Yahaba, Iwate, 028-3609, Japan

Response to Reviewer #1:

1. Since bar graphs were changed to plots, now the data became interpretable. On the other hand, values for average and SD overlapped with plots in the revised figures. For this, the referee asks authors to describe numerical values of average and SD in the main text, so that readers may understand the meaning of data. For example, page 11, second and third paragraphs, sentences for Fig 2 and Fig 3 needs numerical values for each group, neoplasm and healthy groups.

Reply: 

In accordance with your suggestion, we have added numerical values for the average and SD in the Results section (Page 11, Lines 12), (Page 11, Lines 13), (Page 11, Lines 15), (Page 11, Lines 17), (Page 12, Lines 3), (Page 12, Lines 5), (Page 12, Lines 8), (Page 13, Lines 5), (Page 13, Lines 6), (Page 13, Lines 7), (Page 13, Lines 8), (Page 13, Lines 14) and (Page 13, Lines 17).

2. looking at Fig 2A and 2B, one single plot in neoplasm group seem to elevate the average in neoplasm group. They stated in the text that “the number of PB-Muse cell was higher in the group with both benign and malignant neoplasms (n =55) than in the healthy group, which included four liver donors (n = 14, p = 0.056; Fig. 2A).” Since there was no statistical difference, authors should not carelessly state higher. “High but without statistical significance” might be better sentence.

The same issue falls into other “higher” parts throughout the text.

Furthermore, authors should check the appropriateness of the single dot in the neoplastic/metastatic patients in Fig 2 to be included as a proper sample. This single dot seems to distorts the data that it should be.

Reply: 

Thank you very much for your invaluable comments. We have changed the indicated sentences (Page 2, Lines 9), (Page 11, Lines 11), (Page 12, Lines 7), (Page 13, Lines 6), (Page 15, Lines 13) and (Page 16, Lines 7). In the case of the single dot in Fig. 2, that point has been excluded as an outlier because the number of PB-Muse cells before surgery was greater than +/- three times the SDs. We have added explanatory text regarding outliers in the Materials and methods section (Page 8, Lines 1) and Results section (Page 11, Lines 10).

3. page 11, third paragraph: why authors can conclude that POD3 was peak in Fig 3A. There was no statistical significance between POD3 and POD7 (p=ns). The same problem falls onto Fig 3B, as well. How POD7 can be the highest while POD3 and POD7 were ns.

Again a single plot in Fig 3C seems to be extraordinal case and thus is questionable to be included as one of the rational samples.

Reply: 

As suggested by the reviewer, we cannot conclude that POD3 was a peak in Fig 3A and POD7 was the highest in Fig 3B. We have changed the indicated sentence (Page 11, Lines 17) and (Page 12, Lines 2). In the case of the single dot in Fig. 3C, that point has been excluded as an outlier because Max PB-Muse was greater than +/- three times the SDs. We have added explanatory text regarding outliers in the Materials and methods section (Page 8, Lines 1) and Results section (Page 8, Lines 13). We have also changed the data throughout the text after reanalysis.

4. page 13, top line: the MHR group (1,852.0 ± 2,431.8 cells/100μL) was higher than in the mhr group (847.4 ± 1,047.9 cells/100μL, p = 0.051; Fig. 4B -> higher but without statistical significance.

Reply: 

In accordance with your suggestion, we have changed the indicated sentence (Page 13, Lines 6).

5. page 11: “ΔMuse was significantly higher in patients with ΔVolume ≥ 1.19 (16.3 ± 25.2) than in those with ΔVolume ˂1.19 (8.4 ± 26.4, p = 0.043; Fig. 5B). ΔMuse did not correlate with ΔVolume (p = 0.917; Fig. 5C).” -> The two sentences are inconsistent. Needs to be explained or reconsider the statement.

Reply: 

Thank you for providing these insights. Although Fig. 5C showed a correlation, and not a relation between cause and effect, as you pointed out, the two sentences are too confusing. We have therefore removed Fig. 5C and the relevant sentence from the Results section and Discussion sections.

6. Table 2 is sloppy. Extra lines should be deleted.

Reply: 

Thank you for your comment. As you pointed out, this is a mistake. The extra lines have been deleted.

Response to Reviewer #2:

1. Figures 3D/E have to be redone without including the outlier. P values needs to be re-calculated and results and discussion needs to modified accordingly.

Reply: 

Thank you very much for your important comments. A case of Max PB-Muse greater than +/- three times SDs has been excluded as an outlier. We have performed a reanalysis that excluded this case. After this reanalysis, the Max PB-Muse and the ΔMuse did not correlate with the Max AST. Therefore, we have removed Fig. 3D and E and the relevant sentences from the Results section and Discussion sections.

---

## [Decision Letter · Decision Letter 2]

6 Jul 2022

Mobilization of multilineage-differentiating stress-enduring cells into the peripheral blood in liver surgery: A cohort study

PONE-D-22-04405R2

Dear Dr. koji kikuchi,

We’re pleased to inform you that your manuscript has been judged scientifically suitable for publication and will be formally accepted for publication once it meets all outstanding technical requirements.

Kind regards,

Chengming Fan, MD, PhD

Academic Editor

PLOS ONE

Additional Editor Comments (optional):

I think it could be accepted.

Reviewers' comments:

Reviewer's Responses to Questions

**Comments to the Author**

1. If the authors have adequately addressed your comments raised in a previous round of review and you feel that this manuscript is now acceptable for publication, you may indicate that here to bypass the “Comments to the Author” section, enter your conflict of interest statement in the “Confidential to Editor” section, and submit your "Accept" recommendation.

Reviewer #1: All comments have been addressed

Reviewer #2: All comments have been addressed

2. Is the manuscript technically sound, and do the data support the conclusions?

Reviewer #1: Yes

Reviewer #2: Yes

3. Has the statistical analysis been performed appropriately and rigorously? 

Reviewer #1: Yes

Reviewer #2: Yes

4. Have the authors made all data underlying the findings in their manuscript fully available?

Reviewer #1: Yes

Reviewer #2: Yes

5. Is the manuscript presented in an intelligible fashion and written in standard English?

Reviewer #1: Yes

Reviewer #2: Yes

6. Review Comments to the Author

Reviewer #1: The paper has revised substantially. In the figures, the display of p-values is inconsistent. In some figures p value is not mentioned but simply indicated ns, and in other figures, the p-value is displayed numerically. Please unify to either.

Reviewer #2: Authors answered all my questions (results, figures, tables, discussion). Therefore MS is accepted for publication.

7. PLOS authors have the option to publish the peer review history of their article (what does this mean?). If published, this will include your full peer review and any attached files.

Reviewer #1: No

Reviewer #2: No

---

## [Editor Report · Acceptance letter]

12 Jul 2022

PONE-D-22-04405R2 

Mobilization of Multilineage-Differentiating Stress-Enduring Cells into the peripheral blood in Liver Surgery 

Dear Dr. Kikuchi:

I'm pleased to inform you that your manuscript has been deemed suitable for publication in PLOS ONE. Congratulations! Your manuscript is now with our production department. 

Kind regards, 

on behalf of

Dr. Chengming Fan 

Academic Editor

PLOS ONE